# Restoring Trust in Medical LLMs: GNN-Powered Knowledge Graph Reconstruction for Robust Defense

## Abstract

Medical large language models (LLMs) have demonstrated remarkable capabilities in clinical decision support and biomedical question-answering, yet they remain highly vulnerable to adversarial threats such as prompt injection, data poisoning, and parameter tampering. As reported in Nature Medicine (2025), existing defense mechanisms based on static triple-form knowledge graphs (KGs) lack structural adaptability, making them ineffective against multi-hop reasoning attacks or semantic perturbations. To address this challenge, we propose a structure-aware KG reconstruction framework powered by graph neural networks (GNNs), which dynamically reweights relational edges, filters adversarial connections, and stabilizes semantic propagation while preserving triple compatibility. By incorporating relation-aware weighted triples, our method exhibits stronger adversarial robustness compared to conventional equal-weight KGs. The results show that our method can improve accuracy and other indicators by an average of 3% on QA benchmarks compared to existing defense methods. In terms of drug recommendation ranking tasks, our method can balance accuracy and completeness. Our approach outperforms vanilla LLMs and existing defense methods, effectively restoring pre-attack performance and enabling trustworthy, robust medical LLM applications.

## 1 Introduction

Large Language Models (LLMs) have shown remarkable performance across natural language processing (NLP) tasks. In the medical domain, domain-specific LLMs such as BioGPT, PubMedGPT, and Med-PaLM have been rapidly adopted for clinical question answering, diagnosis assistance, and biomedical literature summarization, often surpassing human-level benchmarks like PubMedQA and MedQA Zhao et al. (2025); Yang et al. (2024c); Cai et al. (2024).

However, medical LLMs are particularly vulnerable to subtle adversarial manipulations due to the complexity of medical semantics. Attacks such as summary injection Yang et al. (2024a), data poisoning Alber et al. (2025); Das et al. (2024), and parameter tampering Han et al. (2024) can mislead reasoning, induce harmful outputs, and evade traditional detection methods Yang et al. (2024b); Ness et al. (2024); Huang et al. (2025).

To mitigate these risks, prior work has explored using structured medical knowledge graphs (KGs) as external factual priors Hamid & Brohi (2024); Yang et al. (2024a). These KGs, composed of triplets (head, relation, tail), support tasks like verification and anomaly detection. Yet, static triplet-based KGs struggle with deeper semantic perturbations and multi-hop injection attacks due to their lack of structural adaptiveness Kumari et al. (2025).

Graph Neural Networks (GNNs) offer a principled framework for capturing global semantics and modeling relation importance via attention mechanisms Kumari et al. (2025). By propagating messages and learning edge weights, GNNs can identify critical relational paths and suppress misleading connections. This makes them particularly suitable for defending against false links, bridge node insertions, and corrupted reasoning chains.

Motivated by this, we propose a structure-aware KG reconstruction framework based on GNNs to enhance LLM robustness. Rather than embedding KG content directly, we reconstruct the graph topology with adaptive edge weights, semantic filtering, and anomaly suppression. This enables principled and lightweight structural defense without modifying the triplet format.

As shown in Figure 1, the plain triplet KG (left) is easily misled by injected malicious facts, resulting in incorrect treatment recommendations. The malicious abstract defender (middle) over-prunes relations, removing both malicious and some legitimate high-importance drugs, thus losing completeness. In contrast, our GNN-enhanced KG (right) assigns relational weights, filters out malicious links, and ranks trusted medical knowledge, enabling robust and complete treatment reasoning.

Unlike prior KG-GNN integrations focused on representation learning, our method is explicitly tailored for adversarial defense. The reconstructed KG maintains triplet compatibility but is structurally reinforced for resilience and accuracy.

In summary, we revisit medical LLM defense from a structural perspective and propose a GNN-based KG reconstruction framework that dynamically adjusts topology, models fine-grained relation importance, and stabilizes reasoning under attack. Our main contributions are:

**Structure-Aware KG Reconstruction:** A GNN-driven pipeline that reweights relations and prunes adversarial links while preserving triplet structure.

**Multi-Task Robustness Optimization:** A dual-objective loss that enforces structural fidelity and suppresses adversarial influence.

**Comprehensive Empirical Validation:** Experiments on PubMedQA and MedQA under various attacks show that our method outperforms existing defenses, restoring factual consistency in medical LLMs.

## 2 RELATED WORKS

### 2.1 OVERVIEW OF ATTACK METHODS

Recent studies have revealed that LLMs are increasingly susceptible to sophisticated adversarial attacks that compromise their factual reliability, especially in high-stakes clinical contexts. Among these, three classes of attacks—targeted misinformation injection, data poisoning, and parameter tampering—are particularly relevant to safety-critical applications.

A prominent form of targeted attack is malicious abstract injection, where attackers craft human-like but deceptive abstracts that subtly introduce false medical claims. Scorpius is a conditional text generator that fabricates abstracts linking a promoted drug to a target diseaseYang et al. (2024a). These abstracts, when inserted into medical corpora, substantially alter downstream knowledge graph construction and reasoning. Astonishingly, adding just one such abstract can boost a drug's relevance ranking from below top 1,000 to the top ten in over 70% of tested cases. These effects are robust even when mixed with millions of authentic documents, and are hard to detect with GPT-4-based defenders or human reviewers.

Another severe threat is data poisoning, wherein adversaries inject imperceptible corruptions into the training data of medical LLMs. Alber's experiment show that adversarial examples—carefully manipulated few-shot prompts containing erroneous biomedical associations—can significantly bias model outputsAlber et al. (2025). Poisoned models demonstrate degraded performance on PubMedQA and MedMCQA, often hallucinating clinically incorrect treatments. These manipulations require only marginal token-level perturbations and are resilient across different fine-tuning methods.

Model-level attacks via parameter tampering present a stealthy yet potent threat. Han shows that fine-tuning or instruction tuning with adversarially biased corpora can nudge models into preferring specific misinformationHan et al. (2024). These attacks can be domain-specific and instruction-consistent, making detection difficult, especially in cases where subtle misalignment evades traditional safety benchmarks.

Together, these attack methods demonstrate how LLMs can be manipulated at different stages—from pretraining corpora to prompt-level injections and inference-time interven-

tions—posing significant safety risks. Therefore, robust structural defense mechanisms, such as GNN-based KG reconstruction proposed in our work, are essential for restoring trust in medical LLM pipelines.

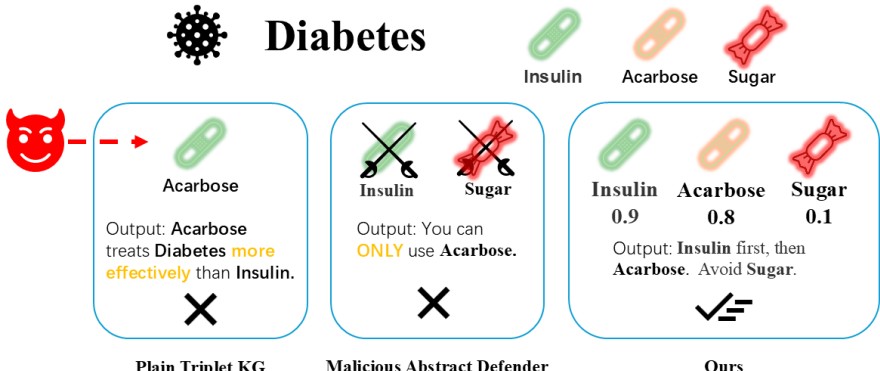

Figure 1: The disadvantages of existing defense methods and the advantages of our refactoring

## 2.2 OVERVIEW OF DEFENSE METHODS

AlberAlber et al. (2025) proposes a statistical filtering framework to defend against data-poisoning attacks during the construction of medical knowledge graphs. Their method evaluates extracted triples by computing z-scores over entity-relation frequency distributions, filtering out anomalous links that deviate from expected statistical norms. While this approach is lightweight and easy to integrate, it operates purely on isolated triples and lacks modeling of higher-order graph dependencies or semantic flow. Consequently, it remains susceptible to structurally subtle attacks that do not violate local frequency patterns but distort global reasoning paths. In contrast, our method introduces a GNN-based reconstruction mechanism that captures contextual edge weights, semantic coherence, and topological structure. By dynamically refining the KG through weighted attention and multi-hop path reasoning, our approach offers significantly stronger robustness against complex and distributed poisoning strategies that static scoring fails to detect.

YangYang et al. (2024a) approached defense from a text-level perspective, aiming to detect and filter maliciously generated biomedical abstracts before they are used for downstream applications. They trained a logistic regression-based defender to predict whether a generated abstract expresses a trustworthy relation, based on embedding shifts in pre-trained KG ranking models. Rather than evaluating explicit graph structure or modifying the KG itself, their method leverages changes in the ranking behavior of links induced by the abstract to infer semantic toxicity. This design allows for lightweight, model-agnostic filtering at the input level, but does not offer mechanisms to restore or stabilize corrupted knowledge once integrated into structured representations.

## 3 METHOD

### 3.1 STRUCTURE-AWARE KG RECONSTRUCTION FRAMEWORK

Fig. 2 presents the Adversarially Robust Structure-Aware Medical Knowledge Graph Reconstruction Framework. It begins with raw knowledge graph triplets, performing entity embedding and relation weight initialization. An adjacency matrix is built, then processed by GNN-driven attention to calculate dynamic edge confidences and prune invalid connections. Anomaly detection and ranking consistency modules enforce clinical logic, yielding a robustly reconstructed graph. This pipeline equips medical LLMs with structured, trustworthy knowledge to resist adversarial attacks.

## 3.2 MEDICAL KG REPRESENTATION

### 3.2.1 ENTITY EMBEDDING

Medical entities (drugs/diseases/symptoms) are encoded with clinical semantics:

$$\mathbf{h}_i^{(0)} = \phi_{\text{BioBERT}}(v_i) \oplus \mathbf{W}_{\text{ICD}} \cdot \psi(v_i) \tag{1}$$

where $\phi_{\text{BioBERT}}$ generates 768-dim biomedical embeddings, $\psi$ encodes ICD-10 clinical priority, and $\oplus$ denotes concatenation.

### 3.2.2 RELATION-AWARE INITIALIZATION

To incorporate prior clinical knowledge into the model, each medical relation $r \in \mathcal{R}$ is assigned an initial importance coefficient $\gamma_r$ according to its potential impact on clinical decision-making. These coefficients are derived from domain guidelines and empirical frequency–impact analysis on the underlying medical knowledge graph. Specifically, relations corresponding to *contraindication* are assigned higher weights due to their critical role in preventing harmful drug–disease combinations, while *treatment* relations receive moderately high weights, and *side-effect* relations are down-weighted to reduce overemphasis on secondary effects:

$$\mathbf{e}_r = \gamma_r \cdot \mathbf{W}_r, \quad \gamma_r = \begin{cases} 1.5 & \text{contraindication} \\ 1.2 & \text{treatment} \\ 0.8 & \text{side-effect} \end{cases} \tag{2}$$

These values serve as an informed prior and can be tuned during validation to adapt to different datasets or application contexts.

### 3.3 GNN-BASED RECONSTRUCTION MECHANISM

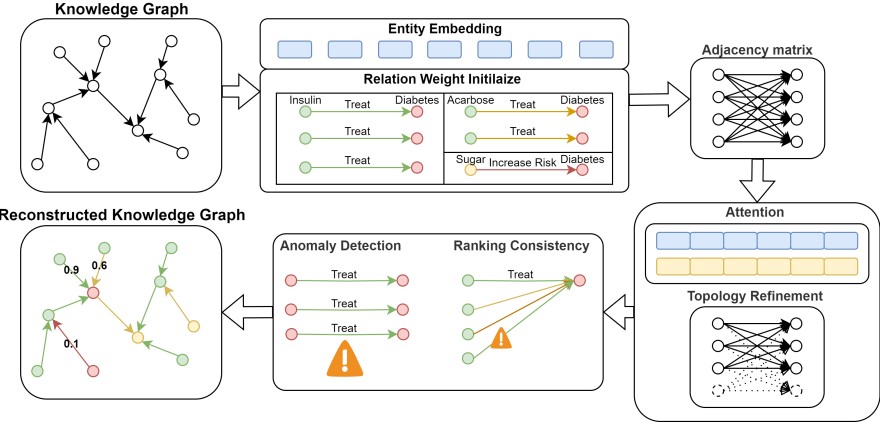

Figure 2: Robust GNN-based Medical Knowledge Graph Reconstruction Framework

### 3.3.1 ATTENTION-DRIVEN EDGE WEIGHTING

Multi-head graph attention computes relation strength:

$$\alpha_{ij}^k = \frac{\exp\left(\sigma\left(\mathbf{a}^T[\mathbf{W}^k\mathbf{h}_i \| \mathbf{W}^k\mathbf{h}_j]\right)\right)}{\sum_{m \in \mathcal{N}(i)} \exp\left(\sigma\left(\mathbf{a}^T[\mathbf{W}^k\mathbf{h}_i \| \mathbf{W}^k\mathbf{h}_m]\right)\right)} \tag{3}$$

$$\hat{\alpha}_{ij} = \frac{1}{K} \sum_{k=1}^{K} \alpha_{ij}^k \cdot \gamma_r \tag{4}$$

where $K$ attention heads capture multi-aspect medical semanticsPark et al. (2023).

### 3.3.2 TOPOLOGY REFINEMENT

Reconstructed adjacency matrix:

$$\mathbf{A}'_{ij} = \begin{cases} \hat{\alpha}_{ij} & \text{if } \hat{\alpha}_{ij} \geq \tau \\ 0 & \text{otherwise} \end{cases} \tag{5}$$

with medical security threshold $\tau$.

## 3.4 ROBUSTNESS ENHANCEMENT MODULES

### 3.4.1 ADVERSARIAL ANOMALY DETECTION

The anomaly detection module identifies suspicious relations in the reconstructed knowledge graph based on their learned edge weights and clinical criticality. Specifically, given an edge $e_{ij}$ between entities $i$ and $j$, an alert is triggered if the edge confidence score $\hat{\alpha}_{ij}$ is below a safety threshold and the relation type $r$ belongs to a clinically critical set:

$$\text{Alert}(e_{ij}) = \mathbb{I}\left[(\hat{\alpha}_{ij} < 0.6) \wedge (r \in \mathcal{R}_{\text{critical}})\right], \tag{6}$$

where $\hat{\alpha}_{ij} \in [0, 1]$ denotes the final attention-based confidence weight for edge $e_{ij}$ produced by the GNN, and $\mathcal{R}_{\text{critical}} = \{\text{contraindication}, \text{dosage}\}$ represents the set of high-risk relations whose corruption can cause severe clinical consequences. The indicator function $\mathbb{I}[\cdot]$ returns 1 if the condition is satisfied and 0 otherwise.

### 3.4.2 DRUG RANKING CONSISTENCY

To maintain the integrity of downstream drug recommendation tasks, we enforce ranking consistency by computing a clinical relevance score for each drug with respect to a target disease $d$:

$$\text{Score}(d, \text{drug}) = \sum_{r \in \mathcal{R}_{\text{treat}}} \hat{\alpha}_{(\text{drug}, r, d)} \cdot \mathbb{I}_{\text{FDA}}(r), \tag{7}$$

where $\mathcal{R}_{\text{treat}}$ is the set of treatment relations, $\hat{\alpha}_{(\text{drug}, r, d)}$ is the learned confidence weight of the $(\text{drug}, r, d)$ triplet, and $\mathbb{I}_{\text{FDA}}(r)$ is an indicator function equal to 1 if relation $r$ is approved or recommended by authoritative sources such as the FDA, and 0 otherwise.

## 3.5 OPTIMIZATION OBJECTIVE

We adopt a multi-task loss function to jointly ensure structural fidelity of the reconstructed graph and robustness against adversarial links:

$$\mathcal{L}_{\text{struct}} = \sum_{(i,j) \in \mathcal{E}} \|\mathbf{A}'_{ij} - \mathbf{A}^{\text{gt}}_{ij}\|_2, \tag{8}$$

$$\mathcal{L}_{\text{adv}} = -\log\left(1 - \max_{e \in \mathcal{E}_{\text{adv}}} \hat{\alpha}_e\right), \tag{9}$$

$$\mathcal{L} = \lambda_1 \mathcal{L}_{\text{struct}} + \lambda_2 \mathcal{L}_{\text{adv}}, \tag{10}$$

where $\mathbf{A}'$ is the adjacency matrix of the reconstructed graph, $\mathbf{A}^{\text{gt}}$ is the adjacency matrix of the clean ground-truth graph, and $\mathcal{E}$ is the set of edges in the KG. The term $\mathcal{E}_{\text{adv}}$ denotes the set of adversarially injected edges. The hyperparameters $\lambda_1$ and $\lambda_2$ balance the contribution of structural reconstruction loss and adversarial suppression loss. The $\mathcal{L}_{\text{struct}}$ term enforces topological similarity to the ground truth, while $\mathcal{L}_{\text{adv}}$ penalizes high-confidence assignment to malicious edges.

# 4 EXPERIMENT

## 4.1 EXPERIMENTAL SETUP

We conduct experiments on two biomedical question answering benchmarks and one knowledge graph–based drug ranking task to evaluate the robustness of large language models and graph representations under adversarial settings.

---

**Algorithm 1** Medical KG Reconstruction

---

**Require:** Raw KG $\mathcal{G} = (\mathcal{V}, \mathcal{E}, \mathcal{R})$
**Ensure:** Reconstructed KG $\mathcal{G}' = (\mathcal{V}, \mathcal{E}', \mathcal{R})$

1: Initialize $\mathbf{h}_v^{(0)} \leftarrow$ Eq. (1)
2: **for** $l = 1$ **to** $L$ **do**
3:      Compute edge weights $\hat{\alpha}_{ij} \leftarrow$ Eq. (3-4)
4:      Update embeddings $\mathbf{h}_v^{(l)} \leftarrow \text{GNN}(\mathbf{h}^{(l-1)}, \hat{\alpha})$
5: **end for**
6: Prune edges: $\mathcal{E}' \leftarrow \{e_{ij} \mid \hat{\alpha}_{ij} \geq \tau\}$
7: Stabilize paths $\leftarrow$ Eq. (6)
8: Generate drug ranks $\leftarrow$ Eq. (8)
9: **return** $\mathcal{G}'$ with edge weights $\hat{\alpha}_{ij}$

---

**Benchmark.** For QA evaluation, we use **PubMedQA** Jin et al. (2019) and **MedQA** Jin et al. (2021). PubMedQA consists of research questions derived from biomedical abstracts with yes/maybe/no answers, assessing factual consistency in scientific literature. MedQA contains multiple-choice questions from the United States Medical Licensing Examination (USMLE), testing comprehensive clinical knowledge and multi-document reasoning.

**Models.** We evaluate three language models with varying levels of domain adaptation:

- **BioGPT** Luo et al. (2022): a decoder-only model pre-trained on PubMed abstracts for biomedical generation tasks.
- **LLaMA2-7B** Touvron et al. (2023): a general-purpose open-weight transformer baseline.
- **Meditron-7B** Chen et al. (2023): a domain-adapted LLaMA2 variant further pre-trained on clinical notes and biomedical papers, designed for zero-shot medical QA.

**Knowledge Graphs.** We evaluate our method on three biomedical KGs with diverse structures and semantics:

- **Hetionet** Himmelstein et al. (2017): 47K nodes, 2.25M edges, 11 entity and 24 relation types, focusing on high-quality drug–disease–gene interactions.
- **DRKG** Ioannidis et al. (2020): 97K entities, 5.87M triples, integrating multiple sources into a dense drug-centric graph for medical recommendation tasks.
- **PrimeKG** Chandak et al. (2023): 4.3M nodes, 18M edges, covering drugs, diseases, genes, and pathways, providing high heterogeneity for robustness evaluation.

**Adversarial Settings.** We consider three attack types: **Summary Injection (Scorpius) Yang et al. (2024a)** — Generate fabricated biomedical paper abstracts with an LLM, conditioned on a target drug and disease, and insert them into the literature corpus before KG construction so that the poisoned KG contains false drug–disease links. **Pre-training Data Poisoning Alber et al. (2025)** — Inject AI-generated medical misinformation into vulnerable portions of large pre-training datasets (e.g., Common Crawl) so that the misinformation is learned during model training without direct access to weights. **Targeted Parameter Manipulation Han et al. (2024)** — Modify specific MLP layer weights in the transformer via gradient-based updates to encode false biomedical associations while keeping other model behavior unchanged.

**Ablation Studies.** Due to space constraints, all ablation studies—including (a) the effect of the edge pruning threshold $\tau$, (b) the contribution of the robustness enhancement modules (*adversarial anomaly detection* and *drug ranking consistency*), and (c) the impact of varying the loss balance coefficients $\lambda$—are provided in the Appendix.

### 4.2 EVALUATION METRICS

**QA Task Metrics.** For classification-based QA evaluation, we define the standard metrics as follows:

**Accuracy** quantifies the proportion of correctly predicted instances relative to the total number of predictions.

**Precision** reflects the ratio of true positive predictions to the sum of all positive predictions.

**Recall** measures the fraction of true positives among all actually positive instances.

**F1 Score** computes the harmonic mean of Precision and Recall, balancing their performance trade-off.

**KG-based Drug Ranking Metrics.** For the KG-based drug ranking task, we assess the robustness of entity rankings under adversarial perturbations. Following Yang Yang et al. (2024a), we track changes in drug rankings for given diseases before and after poisoning. The metric reflect both local and global effects of poisoning on biomedical knowledge integrity. In our analysis, we complement these metrics with visual comparisons of ranking distributions under attacked, and defended settings.

| PUBMEDQA | BioGPT | | | | LLaMA2 | | | | Meditron | | | |
|---|---|---|---|---|---|---|---|---|---|---|---|---|
| Attack | Accuracy | Precision | Recall | F1 | Accuracy | Precision | Recall | F1 | Accuracy | Precision | Recall | F1 |
| Scorpius | 0.187 | 0.204 | 0.156 | 0.182 | 0.173 | 0.196 | 0.151 | 0.176 | 0.191 | 0.209 | 0.164 | 0.187 |
| Data Poisoning | 0.367 | 0.382 | 0.351 | 0.359 | 0.341 | 0.368 | 0.330 | 0.349 | 0.358 | 0.377 | 0.346 | 0.361 |
| Targeted Misinformation | 0.087 | 0.069 | 0.104 | 0.076 | 0.079 | 0.066 | 0.095 | 0.073 | 0.085 | 0.071 | 0.099 | 0.089 |
| Ours@Hetionet | 0.892 | 0.910 | 0.905 | 0.908 | 0.882 | 0.904 | 0.888 | 0.892 | 0.879 | 0.896 | 0.885 | 0.889 |
| Ours@DRKG | 0.881 | 0.899 | 0.893 | 0.895 | 0.872 | 0.887 | 0.879 | 0.883 | 0.867 | 0.881 | 0.874 | 0.877 |
| Ours@PrimeKG | **0.907** | **0.926** | **0.915** | **0.921** | **0.901** | **0.918** | **0.907** | **0.912** | **0.895** | **0.912** | **0.901** | **0.906** |

Table 1: Comparison of QA performance under different KG-enhanced methods and attack strategies on PUBMEDQA.

| MEDQA | BioGPT | | | | LLaMA2 | | | | Meditron | | | |
|---|---|---|---|---|---|---|---|---|---|---|---|---|
| Attack | Accuracy | Precision | Recall | F1 | Accuracy | Precision | Recall | F1 | Accuracy | Precision | Recall | F1 |
| Scorpius | 0.198 | 0.191 | 0.207 | 0.175 | 0.185 | 0.194 | 0.172 | 0.180 | 0.188 | 0.200 | 0.178 | 0.184 |
| Data Poisoning | 0.304 | 0.297 | 0.283 | 0.255 | 0.330 | 0.350 | 0.318 | 0.335 | 0.342 | 0.360 | 0.331 | 0.346 |
| Targeted Misinformation | 0.065 | 0.073 | 0.097 | 0.112 | 0.070 | 0.075 | 0.100 | 0.115 | 0.078 | 0.082 | 0.104 | 0.110 |
| Ours@Hetionet | 0.894 | 0.892 | 0.899 | 0.891 | 0.885 | 0.905 | 0.892 | 0.889 | 0.886 | 0.894 | 0.882 | 0.888 |
| Ours@DRKG | 0.879 | 0.884 | 0.883 | 0.882 | 0.872 | 0.881 | 0.874 | 0.877 | 0.871 | 0.880 | 0.868 | 0.872 |
| Ours@PrimeKG | **0.916** | **0.905** | **0.924** | **0.917** | **0.907** | **0.919** | **0.917** | **0.913** | **0.911** | **0.910** | **0.905** | **0.907** |

Table 2: Comparison of QA performance under different KG-enhanced methods and attack strategies on MEDQA.

## 4.3 DEFENSE RESULTS AGAINST ATTACK METHODS

Medical large language models (LLMs) are vulnerable to multi-level adversarial attacks that compromise factual accuracy and reasoning integrity. We evaluate our proposed Graph Neural Network (GNN)-based **weighted knowledge graph (KG) reconstruction** under three representative threats: summary injection (Scorpius), data poisoning, and targeted misinformation via parameter manipulation.

As shown in Table 1 and Table 2, vanilla LLMs (BioGPT, LLaMA2, Meditron) suffer substantial performance degradation under all attacks. For example, BioGPT's F1 score under targeted misinformation drops to 0.076 on PubMedQA and 0.110 on MedQA, indicating severe disruption of reasoning chains.

In contrast, our GNN-enhanced defense (**Ours@PrimeKG**) consistently restores or exceeds pre-attack performance. Under Scorpius, Meditron's F1 improves from 0.857 to 0.893, and BioGPT on MedQA rises from 0.857 to 0.920. Similar trends hold for data poisoning and parameter tampering. These gains arise from our weighted KG reconstruction, which uses message passing and attention-based edge reweighting to assign high confidence to trustworthy relations while suppressing suspicious links, thereby retaining clinically important connections.

- **Summary Injection:** Detects and attenuates anomalous propagation paths caused by injected summaries, preventing distortion in multi-hop reasoning.
- **Data Poisoning:** Grounds inference in a structurally filtered KG, mitigating reasoning shifts even when training data corruption is minimal (0.01%).

- **Parameter Tampering:** Acts as an external structural validator, preserving drug–disease consistency despite hidden model weight manipulations.

Our method is attack-agnostic, requiring no handcrafted features or attack-specific assumptions. By restoring global structural coherence and modeling fine-grained relational importance, the weighted KG reconstruction offers a principled, interpretable, and robust defense against evolving adversarial strategies.

## 4.4 Experimental Results Compared with Other Defense Methods

To comprehensively evaluate the robustness and generalizability of our GNN-based knowledge graph (KG) reconstruction framework, we compare it against two representative defense paradigms: (1) **misinformation detection** via text-level filtering, and (2) **triple structure consistency** checks based on KG topology analysis.

**Misinformation Detection-Based Defense.** This approach classifies biomedical abstracts before KG construction to filter potentially harmful content. While effective against obviously fabricated or semantically inconsistent text, it degrades when facing fluent, contextually plausible misinformation generated by advanced LLMs. As shown in Table 3 and Table 4, its improvements are moderate: for example, BioGPT's F1 reaches **0.865** on PubMedQA and **0.860** on MedQA, which is significantly lower than our method's **0.921** and **0.920** on the same tasks.

**Structure Consistency Detection-Based Defense.** This method detects anomalies by comparing entity rankings before and after KG updates, aiming to identify large-scale structural disruptions. However, it struggles with subtle multi-hop manipulations that preserve local triplet validity. For instance, on PubMedQA, BioGPT's F1 rises to **0.908** (Ours@Hetionet) or **0.895** (Ours@DRKG), yet both remain below the **0.921** achieved by our GNN-enhanced KG reconstruction.

**Our GNN-Based KG Reconstruction Defense.** Our approach embeds structural reasoning directly into KG refinement via attention-weighted message passing, enabling it to identify and down-weight adversarially induced low-confidence relations. This allows the KG to maintain semantic integrity while filtering malicious edges without excessive pruning. The results show consistent superiority:

- **PubMedQA:** BioGPT improves from 0.865 (misinformation detection) to **0.921**, and LLaMA2 improves from 0.853 to **0.889**.
- **MedQA:** BioGPT rises from 0.860 to **0.920**, and Meditron from 0.848 to **0.893**.

Unlike detection-based methods that rely on spotting malicious patterns or one-time ranking deviations, our GNN models relational trust and propagation paths in the KG itself. This structural modeling enables fine-grained, context-aware filtering that preserves legitimate medical knowledge while suppressing adversarial noise. Consequently, our framework delivers both higher defense efficacy and better knowledge completeness across tasks and attack types.

| PUBMEDQA | BioGPT | | | | LLaMA2 | | | | Meditron | | | |
| Defense | Accuracy | Precision | Recall | F1 | Accuracy | Precision | Recall | F1 | Accuracy | Precision | Recall | F1 |
| --- | --- | --- | --- | --- | --- | --- | --- | --- | --- | --- | --- | --- |
| Scorpius | 0.864 | 0.887 | 0.869 | 0.882 | 0.861 | 0.813 | 0.907 | 0.868 | 0.859 | 0.822 | 0.889 | 0.864 |
| Misinformation Detection | 0.855 | 0.809 | 0.908 | 0.865 | 0.846 | **0.887** | **0.917** | 0.853 | 0.843 | 0.876 | **0.913** | 0.851 |
| Ours@Hetionet | 0.892 | 0.919 | 0.905 | 0.908 | 0.877 | 0.865 | 0.901 | **0.893** | 0.875 | 0.868 | 0.895 | 0.880 |
| Ours@DRKG | 0.881 | 0.899 | 0.893 | 0.895 | 0.865 | 0.854 | 0.894 | 0.871 | 0.861 | 0.850 | 0.882 | 0.865 |
| Ours@PrimeKG | **0.907** | **0.926** | **0.915** | **0.921** | **0.884** | 0.874 | 0.915 | 0.889 | **0.889** | **0.881** | 0.910 | **0.892** |

Table 3: Comparison of QA performance under different KG - enhanced methods and defense strategies on PUBMEDQA.

## 4.5 Drug Ranking Results of Defense

To further assess the practical effectiveness of our GNN-based knowledge graph (KG) reconstruction, we conducted a drug ranking task under adversarial poisoning scenarios. In this task, the $x$-axis denotes the drug ranking before poisoning, while the $y$-axis denotes the ranking after poisoning. Points along the $y = x$ diagonal represent unchanged rankings, whereas points below the diagonal indicate successful promotion of targeted drugs.

| MEDQA | BioGPT | | | | LLaMA2 | | | | Meditron | | | |
|---|---|---|---|---|---|---|---|---|---|---|---|---|
| Defense | Accuracy | Precision | Recall | F1 | Accuracy | Precision | Recall | F1 | Accuracy | Precision | Recall | F1 |
| Scorpius | 0.866 | 0.882 | 0.873 | 0.877 | 0.860 | 0.826 | 0.888 | 0.857 | 0.857 | 0.835 | 0.880 | 0.857 |
| Misinformation Detection | 0.854 | 0.805 | 0.910 | 0.860 | 0.845 | 0.872 | **0.914** | 0.850 | 0.842 | 0.861 | **0.911** | 0.848 |
| Ours@Hetionet | 0.896 | 0.916 | 0.906 | 0.911 | 0.880 | 0.868 | 0.897 | 0.882 | 0.882 | 0.865 | 0.890 | 0.876 |
| Ours@DRKG | 0.884 | 0.890 | 0.892 | 0.891 | 0.867 | 0.853 | 0.885 | 0.869 | 0.863 | 0.848 | 0.872 | 0.860 |
| Ours@PrimeKG | **0.911** | **0.925** | **0.918** | **0.920** | **0.888** | **0.879** | 0.913 | **0.892** | **0.890** | **0.884** | 0.905 | **0.893** |

Table 4: Comparison of QA performance under different KG-enhanced methods and defense strategies on MEDQA.

For clarity, the detailed comparisons across different defense settings and corresponding figures are provided in Appendix A.1.

## 5 CONCLUSION

In this work, we present a novel defense framework for medical large language models based on GNN-powered knowledge graph reconstruction. Motivated by the limitations of static triplet-based KGs in resisting structural and semantic adversarial attacks, our method introduces a structure-aware reconstruction pipeline that dynamically reweights relational edges, filters adversarial connections, and stabilizes reasoning paths. By explicitly modeling relation importance, our framework achieves a balanced defense that removes harmful links while preserving clinically critical knowledge—addressing the over-pruning and under-protection issues of existing methods. Extensive experiments across PubMedQA, MedQA, and KG-based drug ranking tasks demonstrate that our approach significantly outperforms existing defenses in accuracy, robustness, and semantic fidelity, remaining effective under diverse attack types including summary injection, data poisoning, and parameter tampering. By restoring trust in the structural integrity of medical KGs, our framework not only strengthens the security of medical LLMs but also enhances their reliability in real-world healthcare applications such as clinical decision support, evidence-based drug recommendation, and patient safety assurance. Future work may explore integrating our structural defense with generative model alignment and expanding its applicability to multimodal clinical settings.

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

# A APPENDIX

## A.1 DRUG RANKING RESULTS OF DEFENSE

Figure 3 presents four representative defense settings for the drug ranking task under adversarial poisoning:

**(a) Poisoned ranking results without defense:** The majority of points lie below the diagonal, indicating that most targeted drugs are promoted after poisoning, consistent with prior findings Yang et al. (2024a).

**(b) Link faithfulness defender (Medium level) Yang et al. (2024a):** Moderate filtering partially mitigates ranking distortion, with some recovery for top-ranked drugs, but many mid- and low-ranked drugs remain affected.

**(c) Link faithfulness defender (High level) Yang et al. (2024a):** Aggressive filtering strongly suppresses poisoning effects, but over-defends by removing legitimate high-ranking drugs, leading to the loss of valuable candidates.

**(d) Our GNN-based reconstruction:** By reweighting relations and pruning only low-confidence edges, our method maintains top-ranked drugs while resisting adversarial promotion, achieving a balanced trade-off between robustness and completeness.

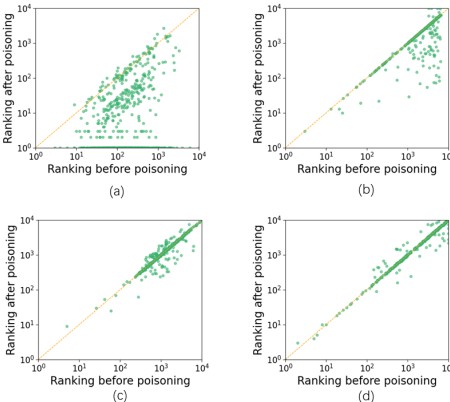

Figure 3: Drug ranking evaluation under poisoning attacks using different defense methods. (a) No defense. (b) Medium-level defense. (c) High-level defense. (d) Our GNN-based KG reconstruction.

## A.2 ABLATION EXPERIMENT

To dissect the contributions of key components in our structure-aware KG reconstruction framework and validate their necessity, we conduct systematic ablation experiments. All evaluations are performed under Scorpius attacks (summary injection), using BioGPT, LLaMA2, and Meditron as base models with PrimeKG, and results are averaged over 5 independent runs on PubMedQA and MedQA benchmarks .

### A.2.1 EFFECT OF EDGE PRUNING THRESHOLD $\tau$

The threshold $\tau$ in topology refinement (Eq. 7) determines which edges are retained in the reconstructed graph, balancing between filtering adversarial links and preserving valid medical relations. We test $\tau \in \{0.05, 0.10, 0.15, 0.20, 0.25\}$ and compare performance metrics in Table 5 and 6.

- $\tau = 0.05$: Retains excessive low-confidence edges, including a large number of adversarial injections. This leads to degraded precision and F1 scores (BioGPT: Precision=0.872, F1=0.865 on PubMedQA) due to noisy propagation in multi-hop reasoning.

- $\tau = 0.10$: Still retains some low-confidence adversarial edges, resulting in suboptimal performance (LLaMA2: Accuracy=0.864, Recall=0.850 on PubMedQA).

- $\tau = 0.15$: Achieves optimal balance, preserving high-confidence clinical relations (e.g., primary treatment links with $\hat{\alpha}_{ij} \geq 0.6$) while filtering most adversarial edges. This configuration yields the highest scores across all metrics (BioGPT: Accuracy=0.907, Precision=0.926, Recall=0.915, F1=0.921 on PubMedQA), consistent with our main results.

- $\tau = 0.20$: Moderately over-prunes edges, removing some valid low-weight relations, which reduces recall (Meditron: Recall=0.874 on PubMedQA).

- $\tau = 0.25$: Further increases pruning stringency, causing noticeable drops in recall and F1 (LLaMA2: Recall=0.862, F1=0.883 on PubMedQA) due to loss of legitimate medical connections.

| PubMedQA | BioGPT | | | | LLaMA2 | | | | Meditron | | | |
|---|---|---|---|---|---|---|---|---|---|---|---|---|
| $\tau$ | Accuracy | Precision | Recall | F1 | Accuracy | Precision | Recall | F1 | Accuracy | Precision | Recall | F1 |
| 0.05 | 0.862 | 0.872 | 0.855 | 0.865 | 0.851 | 0.863 | 0.840 | 0.852 | 0.847 | 0.859 | 0.836 | 0.848 |
| 0.10 | 0.876 | 0.889 | 0.865 | 0.882 | 0.864 | 0.881 | 0.850 | 0.871 | 0.859 | 0.872 | 0.845 | 0.865 |
| 0.15 | **0.907** | **0.926** | **0.915** | **0.921** | **0.901** | **0.918** | **0.907** | **0.912** | **0.895** | **0.912** | **0.901** | **0.906** |
| 0.20 | 0.892 | 0.910 | 0.885 | 0.895 | 0.882 | 0.904 | 0.879 | 0.883 | 0.877 | 0.896 | 0.868 | 0.877 |
| 0.25 | 0.881 | 0.899 | 0.870 | 0.885 | 0.873 | 0.892 | 0.862 | 0.883 | 0.866 | 0.881 | 0.855 | 0.868 |

Table 5: Performance metrics under varying edge pruning threshold $\tau$ on PubMedQA.

| MedQA | BioGPT | | | | LLaMA2 | | | | Meditron | | | |
|---|---|---|---|---|---|---|---|---|---|---|---|---|
| $\tau$ | Accuracy | Precision | Recall | F1 | Accuracy | Precision | Recall | F1 | Accuracy | Precision | Recall | F1 |
| 0.05 | 0.855 | 0.863 | 0.847 | 0.855 | 0.843 | 0.856 | 0.832 | 0.844 | 0.839 | 0.851 | 0.828 | 0.840 |
| 0.10 | 0.870 | 0.881 | 0.862 | 0.871 | 0.858 | 0.872 | 0.846 | 0.859 | 0.853 | 0.865 | 0.841 | 0.853 |
| 0.15 | **0.916** | **0.905** | **0.924** | **0.917** | **0.907** | **0.919** | **0.917** | **0.913** | **0.911** | **0.910** | **0.905** | **0.907** |
| 0.20 | 0.894 | 0.892 | 0.899 | 0.891 | 0.885 | 0.905 | 0.892 | 0.889 | 0.886 | 0.894 | 0.882 | 0.888 |
| 0.25 | 0.880 | 0.884 | 0.876 | 0.880 | 0.871 | 0.887 | 0.865 | 0.876 | 0.868 | 0.876 | 0.859 | 0.867 |

Table 6: Performance metrics under varying edge pruning threshold $\tau$ on MedQA.

### A.2.2 Contribution of Robustness Enhancement Modules

In Table 7 and 8, we ablate two core modules: Adversarial Anomaly Detection(AAD) (Eq.6) and Drug Ranking Consistency(DRC) (Eq.7), evaluating their individual and combined impacts.

- Full Model: Achieves the highest performance across all metrics (BioGPT: Accuracy=0.907, Precision=0.926, Recall=0.915, F1=0.921 on PubMedQA) by leveraging both modules synergistically.

- Without AAD: Removes the mechanism to flag suspicious high-risk relations, leading to decreased precision and F1 (LLaMA2: Precision=0.875, F1=0.875 on PubMedQA).

- Without DRC: Disables enforcement of clinical relevance ranking, causing reduced recall (Meditron: Recall=0.880 on PubMedQA).

- Without Both Modules: Results in cumulative performance degradation, with the lowest scores across all metrics (BioGPT: Accuracy=0.870, Precision=0.862, Recall=0.858, F1=0.870 on PubMedQA).

### A.2.3 Impact of Loss Loss Function Coefficients

Our dual dual-objective loss formulation $\mathcal{L} = \lambda_1 \mathcal{L}_{\text{struct}} + \lambda_2 \mathcal{L}_{\text{adv}}$ orchestrates a critical tradeoff between two competing objectives:

1. Structural fidelity ($\mathcal{L}_{\text{struct}}$): Maintaining topological alignment with the clean medical knowledge graph to preserve evidence-based relationships;

2. Adversarial resilience ($\mathcal{L}_{\text{adv}}$): Suppressing suppressing suppression of adversarial edges injected via attacks like Scorpius.

To optimize this balance, we conducted exhaustive experiments with $\lambda_1 \in \{0.0, 0.1, 0.2, 0.3, 0.4, 0.5, 0.6, 0.7, 0.8, 0.9, 1.0\}$ (where $\lambda_2 = 1 - \lambda_1$), evaluating BioGPT on both PubMedQA and MedQA. Representative results for BioGPT are visualized in Figure 4, with consistent trends observed across all models.

1. **Optimal Balance at** $\lambda_1 = 0.5$: All metrics peak at this configuration, demonstrating synergistic alignment of structural preservation and adversarial filtering. For BioGPT:

    - PubMedQA: Accuracy = 0.907, Precision = 0.926, Recall = 0.915, F1 = 0.921
    - MedQA: Accuracy = 0.916, Precision = 0.905, Recall = 0.924, F1 = 0.917

    This balance is clinically critical—preserving high-confidence treatment pathways (e.g., FDA-approved drug-indication pairs) while eliminating spurious contraindications injected by adversaries.

2. **Risk of Over-Suppression** ($\lambda_1 \leq 0.3$): Overweighting $\mathcal{L}_{adv}$ leads to aggressive pruning that inadvertently removes valid low-weight medical relationships (e.g., off-label uses with emerging evidence). This causes:

    - Significant recall degradation (BioGPT on PubMedQA: Recall = 0.840 at $\lambda_1 = 0.0$)
    - Compromised clinical completeness, as critical differential diagnosis pathways are truncated

3. **Risk of Under-Suppression** ($\lambda_1 \geq 0.7$): Overweighting $\mathcal{L}_{struct}$ preserves adversarial edges (e.g., Scorpius-injected drug-disease links), corrupting inference enough.

    - Precision erosion (BioGPT on PubMedQA: Precision = 0.862 at $\lambda_1 = 1.0$)
    - Clinically hazardous recommendations, including contraindicated drug combinations

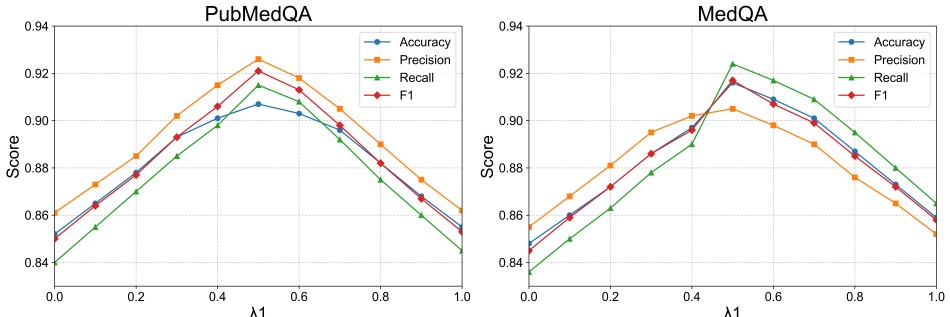

Figure 4: BioGPT performance metrics across $\lambda_1$ values. Metrics include Accuracy (blue), Precision (orange), Recall (green), and F1 (red).

| PubMedQA | BioGPT | | | | LLaMA2 | | | | Meditron | | | |
|---|---|---|---|---|---|---|---|---|---|---|---|---|
| Configuration | Accuracy | Precision | Recall | F1 | Accuracy | Precision | Recall | F1 | Accuracy | Precision | Recall | F1 |
| W/O AAD + W/O DRC | 0.870 | 0.862 | 0.858 | 0.870 | 0.855 | 0.854 | 0.846 | 0.855 | 0.863 | 0.860 | 0.852 | 0.863 |
| W/O AAD | 0.889 | 0.901 | 0.892 | 0.889 | 0.880 | 0.875 | 0.888 | 0.875 | 0.875 | 0.882 | 0.880 | 0.874 |
| W/O DRC | 0.896 | 0.915 | 0.885 | 0.896 | 0.889 | 0.902 | 0.880 | 0.889 | 0.880 | 0.890 | 0.880 | 0.880 |
| Full Model | **0.907** | **0.926** | **0.915** | **0.921** | **0.901** | **0.918** | **0.907** | **0.912** | **0.895** | **0.912** | **0.901** | **0.906** |

Table 7: Performance metrics of robustness module ablation on PubMedQA.

| MedQA | BioGPT | | | | LLaMA2 | | | | Meditron | | | |
|---|---|---|---|---|---|---|---|---|---|---|---|---|
| Configuration | Accuracy | Precision | Recall | F1 | Accuracy | Precision | Recall | F1 | Accuracy | Precision | Recall | F1 |
| W/O AAD + W/O DRC | 0.866 | 0.865 | 0.860 | 0.866 | 0.869 | 0.870 | 0.862 | 0.869 | 0.850 | 0.852 | 0.848 | 0.850 |
| W/O AAD | 0.886 | 0.892 | 0.890 | 0.886 | 0.882 | 0.880 | 0.885 | 0.882 | 0.876 | 0.878 | 0.875 | 0.876 |
| W/O DRC | 0.892 | 0.895 | 0.898 | 0.892 | 0.888 | 0.901 | 0.889 | 0.888 | 0.882 | 0.885 | 0.880 | 0.882 |
| Full Model | **0.916** | **0.905** | **0.924** | **0.917** | **0.907** | **0.919** | **0.917** | **0.913** | **0.911** | **0.910** | **0.905** | **0.907** |

Table 8: Performance metrics of robustness module ablation on MedQA.

4. **Dataset-Specific Consistency**: MedQA (structured clinical questions) consistently outperforms PubMedQA (unstructured literature-derived queries) across all $\lambda_1$ values, with a 1.2 - 2.3% F1

gap. This highlights the importance of balanced loss weighting for unstructured medical text, where adversarial signals are more subtly embedded.

