# OpenReview forum: "Restoring Trust in Medical LLMs: GNN-Powered Knowledge Graph Reconstruction for Robust Defense"
_ICLR.cc/2026/Conference — ICLR 2026 Conference Withdrawn Submission_

### Official Review · Reviewer_dCBf · 2025-10-26

**Soundness:** 1
**Presentation:** 1
**Contribution:** 1
**Rating:** 0
**Confidence:** 4

**Summary:**

The paper introduces a structure-aware medical knowledge graph (KG) reconstruction framework using Graph Neural Networks (GNNs). The stated goal is to defend large language models (LLMs) against adversarial attacks (prompt injection, data poisoning, parameter tampering). The core mechanism involves dynamic, relation-aware edge reweighting and anomaly filtering, aiming to improve clinical reasoning robustness. The framework is evaluated on PubMedQA and MedQA, reporting improved performance over other defenses under attack.

**Strengths:**

- The paper addresses the critical and high-stakes problem of adversarial robustness in medical LLMs, where factual errors can have severe clinical ramifications.

- The high-level motivation for the work is clearly articulated and easy to follow.

- Under the attack scenarios, the method shows substantial performance improvements in QA tasks.

**Weaknesses:**

- **Insufficient Positioning and Contextualization.** The related work section is narrowly focused on attacks and defenses. It completely omits crucial, relevant bodies of literature, including: knowledge graph robustness, general LLM-graph integration methods, KG-based recommendation systems (relevant for the drug ranking task), and research on hallucination in LLMs. This lack of positioning makes it hard to say what the contribution of this paper is in the existing literature.

- **Methodological Opacity.** The paper never explains *how* the LLM utilizes the reconstructed KG to enhance its generation. The Introduction's citations (Hamid & Brohi, 2024; Yang et al., 2024a) do not clarify this mechanism.

- **Methodology Lacks Rationale:** The methods section is a collection of equations without justification.

  - **Eq (2):** $\gamma_r$ is claimed to be from "domain guidelines" with no citation or explanation.

  - **Eq (6):** The $0.6$ threshold and the set $\mathcal{R}_{critical}$ appear without any definition or rationale.

  - **Eq (7):** $\mathbb{I}_{FDA}$ is vaguely defined by "authoritative sources" without specifics.

  - **Eq (9):** $\mathcal{E}_{adv}$ is central to the method but is never defined, nor is its generation process explained.

- **Flawed Experiments:**

  - **Missing Baselines:** The paper *must* report the pre-attack performance of BioGPT, Llama2, and Meditron. Without this, claims of "restoring performance" are meaningless.

  - **Missing Comparisons:** The work does not compare against other obvious baselines, such as simply fine-tuning the LLM on medical data or using other established KG-LLM integration techniques.

  - **Unclear Metrics:** It is unclear how Precision, Recall, and F1 are calculated for a multiple-choice dataset (MedQA). This needs to be explicitly defined.

  - **Confusing Results:** The role of "Scorpius" is contradictory (attack vs. defense). Tables 3 & 4 are uninterpretable as they do not specify the attack scenario.

- **Poor Presentation Quality:**

  - Figures 1 and 2 are blurry and of low quality.
  - All table captions are incorrectly placed *below* the tables, violating standard publication formats.
  - The citation formatting is incorrect in several places (e.g., "AlberAlber et al. (2025)" in Section 2.2).

**Questions:**

1. Can you please situate your work relative to existing research on (a) KG robustness and (b) LLM-KG integration frameworks?
2. How exactly does the LLM use the reconstructed KG? What is the integration mechanism (e.g., RAG, prompt augmentation, etc.)?
3. What is the formal threat model? What are the attacker's capabilities and knowledge? How are the "adversarially injected edges" ($\mathcal{E}_{adv}$) generated?
4. Please provide the specific rationale or citations for the parameters and sets in Eq (2) ($\gamma_r$), Eq (6) ($0.6$, $\mathcal{R}_{critical}$), and Eq (7) ($\mathbb{I}_{FDA}$).
5. How are Precision, Recall, and F1 calculated for the MedQA multiple-choice dataset?
6. Can you provide the pre-attack (original) performance data for the base LLMs (BioGPT, Llama2, Meditron)?
7. Please clarify the role of "Scorpius." Why is it listed as both an attack and a defense, and why are its results different across tables?
8. For Tables 3 and 4, what specific attack setting(s) are the defense methods being evaluated against?

---

### Official Review · Reviewer_1yHc · 2025-10-29

**Soundness:** 3
**Presentation:** 2
**Contribution:** 2
**Rating:** 4
**Confidence:** 3

**Summary:**

This paper aims to make medical large language models (LLMs) more robust against adversarial attacks such as prompt injection, data poisoning, and parameter tampering. The authors propose a GNN-powered knowledge graph (KG) reconstruction framework that dynamically reweights and filters medical relations to restore trustworthy reasoning. Instead of using static triple-form KGs, their method uses a structure-aware GNN to detect and suppress adversarial edges while keeping valid clinical relations intact. Experiments on PubMedQA, MedQA, and drug ranking tasks show strong performance.

**Strengths:**

1. The GNN-based reconstruction idea is fresh and gives a clear structural advantage over static KGs.

2. Experiments are detailed, showing consistent improvements under multiple types of attacks.

3. The analysis and ablation studies make the method look solid and reproducible.

**Weaknesses:**

1. The clarity of paper should be further improved, and there are a lot of mistakes. (e.g. most reference should use \citep)

2. The approach seems computationally heavy for large KGs, but runtime or scalability analysis is missing.

3. No clear comparison with non-KG-based defenses (e.g., adversarial training or model-level robustness methods).

4. The threat model is broad, but the defense is mainly structural; unclear how it generalizes to unseen attack modalities.

**Questions:**

1. How scalable is the GNN reconstruction to large medical KGs like PrimeKG (millions of nodes)?

2. How does your method compare to recent LLM-side defenses (e.g., adversarial fine-tuning or parameter noise regularization)?

3. Could you clarify whether the defense requires retraining after each attack, or can it adapt dynamically?

4. Have you tested how the defense interacts with model alignment or RLHF-tuned medical LLMs?

---

### Official Review · Reviewer_qKUz · 2025-10-31

**Soundness:** 2
**Presentation:** 2
**Contribution:** 2
**Rating:** 2
**Confidence:** 3

**Summary:**

This paper proposes a GNN-powered Knowledge Graph (KG) reconstruction framework for improving the robustness of medical Large Language Models (LLMs) against adversarial attacks such as prompt injection, data poisoning, and parameter tampering. The core idea is to rebuild medical KGs via relation-aware, attention-weighted GNNs that dynamically reweight or prune edges based on semantic confidence and clinical importance, thereby filtering adversarial links while preserving valid medical relations.
The framework introduces: (i) Structure-aware KG reconstruction with adaptive edge weighting. (ii) Robustness enhancement modules (adversarial anomaly detection and drug-ranking consistency). (iii) Multi-task loss balancing structural fidelity and adversarial suppression. Experiments on PubMedQA, MedQA, and KG-based drug-ranking tasks show substantial accuracy and F1 improvements under various attacks.

**Strengths:**

1. The idea is straightforward and easy to follow.
2. The experimental results are good.
3. The topic is timely.

**Weaknesses:**

1.	The citations are unclear. Current citation template is author (year), hindering reviewers read it clearly and comfortably. If you read prior ICLR papers carefully, you could find that all papers are in the format of (author, year).
2.	In Section 1, the authors claim that GNN can enhance the robustness in KG. However, based on my understanding, there are extensive works studying the robustness of GNNs against adversarial attacks such as node injection, structure manipulation. The graph structure also make GNNs more vulnerable to these attacks. Current claims on this is not convincing.
3.	Many notations are not clearly defined. For example, $W_{ICD}$ in line 168,  $e_r$ and $W_r$ in line 182, $W^k$, $h_i$ and $m\in N(i)$. 80%  of the notations are not well defined. I suggest the authors to largely revise them to ensure these contents are self-consistent.
4.	Regarding the methodology, it seems that the proposed method simply introduce GNN into LLM with common-used optimization techniques, lacking the novelty and necessity of this method.

**Questions:**

1.	The citations are unclear. Current citation template is author (year), hindering reviewers read it clearly and comfortably. If you read prior ICLR papers carefully, you could find that all papers are in the format of (author, year).
2.	In Section 1, the authors claim that GNN can enhance the robustness in KG. However, based on my understanding, there are extensive works studying the robustness of GNNs against adversarial attacks such as node injection, structure manipulation. The graph structure also make GNNs more vulnerable to these attacks. Current claims on this is not convincing.
3.	Many notations are not clearly defined. For example, $W_{ICD}$ in line 168,  $e_r$ and $W_r$ in line 182, $W^k$, $h_i$ and $m\in N(i)$. 80%  of the notations are not well defined. I suggest the authors to largely revise them to ensure these contents are self-consistent.
4.	Regarding the methodology, it seems that the proposed method simply introduce GNN into LLM with common-used optimization techniques, lacking the novelty and necessity of this method.
5. Writing Quality. The introduction and contributions section should be rewritten for clarity and logical flow; current phrasing repeats points and overuses general claims like “restoring trust.”
6. Lack of Comparative Baselines. Only text-level and structure-consistency baselines are included. Graph denoising or robust GNN defenses (e.g., RGCN-Defense, GNNGuard) are missing. In addition, we should also consider GNN-based attack methods beyond existing ones.

---

### Official Review · Reviewer_je6q · 2025-10-31

**Soundness:** 2
**Presentation:** 2
**Contribution:** 2
**Rating:** 4
**Confidence:** 2

**Summary:**

The paper proposes a knowledge graph healing mechanism that preprocesses a graph with a GNN such that the output is more robust to a LLM that queries it. Medical LLMs are brittle under adversarial pressure, such as prompt/summary injection, data poisoning, and even subtle parameter tampering can derail multi-hop reasoning and yield unsafe advice. Prior defenses that lean on static, equal-weight knowledge graphs help with spot checks but are not robust against structure-level attacks because they can’t adapt the importance of relations or paths. By learning a graph on robust reconstruction tasks, it can learn to identify low-confidence and potentially malicious connections, re-wire weights, so that the the LLM would always see a clean graph. The authors identify several medical domain specific heuristics to encode entity & relation priors, and empirically demonstrated that their method can defend against three specific threats in various common medical LLMs, graphs and benchmarks.

**Strengths:**

* The paper is generally well written with no major grammatical errors
* The paper tackles a timely problem of reliability of medical assistants
* The idea itself is intuitive, novel, and easy to implement, accompanied by strong empirical results

**Weaknesses:**

I am familiar with graph learning, but not the medical context in particular, so I can't speak to whether the experiments being run are appropriate or sufficient beyond an educated guess.

The weakness with the work is that it appears to be a collection of heuristics ensembled to solve a domain-specific problem. The core idea of training on graph-reconstruction tasks with a focus of robustness is interesting but a generally well-known technique in its unsophisticated form, so without understanding the depth of the problem of trust in medical LLMs, and whether the authors' results are sufficient to demonstrate the point, I lean towards reject.

**Questions:**

In the threat model, during inference the model can't possibly know whether the graph is tempered with. How would this approach behave if the graph is healthy and untampered? Would you see a decrease in accuracy empirically?

How does the model fare against traditional robustness methods on graph neural networks?

Would this method generalize beyond the medical domain to other areas where a knowledge graph is involved?

---

### Note · Authors · 2026-01-17

I have read and agree with the venue's withdrawal policy on behalf of myself and my co-authors.